# Association between Health-Related Physical Fitness and Risk of Dyslipidemia in University Staff: A Cross-Sectional Study and a ROC Curve Analysis

**DOI:** 10.3390/nu14010050

**Published:** 2021-12-23

**Authors:** Yuan Zhou, Jing Zhang, Rong-Hua Liu, Qian Xie, Xiao-Long Li, Jian-Gang Chen, Xin-Liang Pan, Bo Ye, Long-Long Liu, Wan-Wan Wang, Liang-Liang Yan, Wen-Xin Wei, Xin-Cheng Jiang

**Affiliations:** 1School of Physical Education, Shaanxi Normal University, Xi’an 710119, China; yuanzhou@snnu.edu.cn (Y.Z.); lrh@snnu.edu.cn (R.-H.L.); qq77@snnu.edu.cn (Q.X.); lixiaolong@snnu.edu.cn (X.-L.L.); boomye@snnu.edu.cn (B.Y.); liulonglong@snnu.edu.cn (L.-L.L.); wanwanwang@snnu.edu.cn (W.-W.W.); liangliangyan@snnu.edu.cn (L.-L.Y.); wwenxin@snnu.edu.cn (W.-X.W.); jiangxincheng@snnu.edu.cn (X.-C.J.); 2School of Physical Education and Sports, Beijing Normal University, Beijing 100875, China; 202131070010@mail.bnu.edu.cn; 3School of Kinesiology, Beijing Sport University, Beijing 100084, China; panxinliang@bsu.edu.cn

**Keywords:** health-related physical fitness, dyslipidemia, cardiovascular disease, body composition, cardiorespiratory fitness, flexibility, lipid accumulation product, waist-to-height ratio, body mass index

## Abstract

Background: This study aimed to assess the relationship between dyslipidemia (DL) risk and health-related physical fitness (HPF) and evaluated the prognostic value of HPF for risk of DL. Methods: A total of 776 university staff members were recruited, of which 407 were females, and 369 males. Blood samples and HPF tests were collected from all participants after 12 h fasting. Results: The prevalence of DL was 41.77% and 51.49% in female and male university staff members, respectively, and there was no significant difference between genders (χ^2^ = 2.687, *p* = 0.101). According to the logistic regression analysis, age, male sex, GLU, hypertension, BMI, BF, WHtR, and LAP were significant risk factors for DL (*p* < 0.05), VCI and, SAR were significant protective factors for DL (*p* < 0.05), and SMI, GS, and VG were not significantly associated with the risk of DL. The area under the receiver-operating characteristic (ROC) curve (AUC) analysis indicated that, LAP (AUC: 0.730, 95CI%: 0.697–0.762), WHtR (AUC: 0.626, 95CI%: 0.590–0.660), and BMI (AUC: 0.599, 95CI%: 0.563–0.634) are valid predictors of DL, and LAP and WHtR perform better than BMI (Z = 8.074, *p* < 0.001) in predicting DL in male and female university staff members. Conclusion: The risk of DL is significantly related to body composition, cardiorespiratory fitness, and flexibility. LAP and WHtR perform better than BMI in predicting risk of DL in male and female university staff members.

## 1. Introduction

Cardiovascular disease (CVD) is the leading cause of mortality worldwide, accounting for 30% of global deaths [1]. It has been reported that the prevalence of CVD in the Chinese population has continued to increase, and CVD mortality has increased from 2.51 to 3.97 million between 1990 and 2016. CVD has become the leading cause of death in China [2]. Studies have shown that dyslipidemia (DL) is a major risk factor for CVD, and the prevalence has shown an increasing trend in China [3,4]. A recent national survey in China showed that the prevalence of DL at 35 years or older was 34.7% [5]. Furthermore, the incidence of DL has been reported to be as high as 34% in the general population, and has become an emerging epidemic, which is an increasing burden on China [3,6].

DL is usually elevated by levels of total cholesterol (TC), triglycerides (TG), and low-density lipoprotein cholesterol (LDL-C), as well as low levels of high-density lipoprotein cholesterol (HDL-C) and has been recognized as an independent modifiable CVD-related risk factor for atherosclerosis, coronary heart disease, and stroke, while playing an important role in CVD [3,7,8]. Consequently, the early screening and controlling of lipid levels can reduce CVD morbidity and mortality and has high social value [5,8]. Previous studies have indicated that the leading risk factors for DL are age, central obesity, smoking, insufficient exercise, increasing body mass index (BMI), and waist circumference (WC) [9]. For this reason, the World Health Organization (WHO) recommends exercising regularly, avoiding smoking, and adhering to a healthy diet to reduce the risk of CVD. Accumulating evidence has demonstrated that exercise and physical fitness have beneficial effects on lipid metabolism, insulin resistance, and endothelial function [10]. A study recommended moderate aerobic exercise at least five times per week, for at least 30 min, and this has been shown to be protective against CVD [11]. In addition to aerobic exercise, muscular strength and resistance training could have a beneficial effect on CVD risk factors [12,13]. Lavie et al.’s [14] research indicated that resistance training for less than one hour per week and twice weekly were both associated with over 30% reductions in the development of hypertriglyceridemia (HTG). Therefore, within reason, exercising regularly may be a simple, cheap, easy, and safe method for the treatment of DL. Exercise has been shown to have positive impacts on the pathogenesis, symptomatology and physical fitness of individuals with DL, and to reduce cholesterol levels [15]. However, before exercise training, a comprehensive assessment of DL patients must be carried out, because the benefits of the same physical exercise are different for each individual.

Health-related physical fitness (HPF) as a component of physical fitness includes body composition, muscular strength and endurance, cardiorespiratory fitness (CRF), and flexibility, and has been shown to be associated with CVD risk factors [16,17]. Studies indicate that individuals with the same BMI could differ in body composition, which affects their vulnerability to CVD [18]. CRF is one of the most important components of HPF. A study showed that CRF was inversely correlated with CVD, and maintaining or improving CRF levels over time, could reduce CVD risk [14]. A prospective cohort study showed that higher levels of CRF during young adulthood were independently associated with reduced HTG risk [19]. In a meta-analysis, the exercise group exhibited elevated levels of CRF and improved levels of serum lipids; their insulin, TG, leptin, fibrinogen and angiotensin levels were lower, and their HDL-C and interleukin-18 (IL-18) levels were higher compared with the control group [20]. Beyond that, some studies have reported that grip strength and flexibility were inversely associated with risk of CVD [21]. Although numerous studies have demonstrated an association between HPF and CVD risk, the association of DL risk with HPF remains unclear. Therefore, the aim of the present study is to explore the relationship between DL risk and HPF, and evaluate the prognostic value of HPF for risk of DL. It is extremely important to carry out self-assessments for the early detection and prevention of DL in university staff members.

## 2. Materials and Methods

### 2.1. Participant Recruitment and Study Design

A total of 875 university staff members were recruited from the community hospital health management center. Inclusion criteria: (1) willingness to participate in the study; (2) completion of the test independently; (3) 25 years old ≤ age ≤ 60 years old. Exclusion criteria: (1) physical dysfunction, including restricted joint movement and chronic inflammation; (2) Free from diabetes, liver, kidney, and infectious diseases; (3) subjects could not complete all tests. PASS version 15.0 (NCSS, LLC. Kaysville, Utah, USA) was used for sample size estimation, and we also used power (1-β) calculation with α = 0.05(two-sided). A sample size of 494 (197 females and 282 males) was required to achieve 90% statistical power. After inspection, of the 875 participants, 99 did not complete all tests. Eventually, 776 university staff members were included in this study 369 males and 407 females. The characteristics of the study subjects are shown in Table 1. 

Firstly, all participants underwent a standardized medical examination including the collection of blood samples and the assessment of blood pressure, weight, height, waist circumference (WC), and body composition. After this, other HPF tests were performed following a light breakfast in the morning and at least 15 min rest to regain their stamina. Participants signed informed consent forms prior to participation in the test, after being informed of the test procedures, method, and possible risks of the study. The Ethics Committee of Shaanxi Normal University approved this study, and the ethical approval code is 202116009.

### 2.2. Anthropometric and HPF Tests

Body weight, height, and WC were measured by three registered nurses experienced in health-related research. HPF indicators were measured by trained research assistants according to the National Physical Fitness Standards Manual.

The body weight and height were measured using an all-in-one machine(Guo Kang. GK 720, ZaoZhuang, China) an electronic weight scale and ultrasonic height sensor whereupon subjects stood barefoot in a designated position and looked straight ahead. Body mass index (BMI) was calculated as body weight (kg) divided by height squared (m^2^). WC was measured with a tape, and waist-to-height ratio (WHtR) was calculated as WC (cm) divided by height (cm).

The HPF test included body composition, grip strength (GS), vertical jump (VG), vital capacity (VC), and sit-and-reach (SAR) tests. Body fat mass (BF) and skeletal muscle mass (SMM) were measured via bioelectric impedance analyses (BiospaceCo. InBody 230, Seoul, Korea) performed while participants were standing with electrodes on the sole of the foot and holding electrodes in their hands. Skeletal muscle mass index (SMI) was determined using SMM (kg) divided by height (m^2^) squared. GS, VG, VC, and SAR tests were performed using a Physical Fitness Test Station (Taishan Sports Technology. Taishan-TA106, Shenzhen, China). Each indicator was assessed via two trials, and the highest value was used. This machine provides automatic collection and storage of data. VCI was calculated as VC (mL) divided by body weight (kg).

### 2.3. Blood Pressure Measurement and Blood Biochemical Assays

After 12 h of fasting, each of the subjects’ blood pressure and blood biochemical assays were measured by a nurse during health checks. Before taking blood pressure measurements, the participants were required to rest for 5 to 10 min, and the test was performed using an automated blood pressure measurement device (OMRON. HEM-1000, Dalian, China). The automatic analyzer (Beckman Coulter. AU480, California, USA) was used to analyze blood biochemical features. Triglycerides (TG), total cholesterol (TC), high-density lipoprotein cholesterol (HDL-C), low-density lipoprotein cholesterol (LDL-C), and blood glucose (GLU) were determined using enzymatic techniques. Lipid accumulation product (LAP) was calculated as [WC (cm) −58] × [ TG (mmol/L)] for females and [WC (cm) −65] × [TG (mmol/L)] for males [22].

### 2.4. Diagnostic Criteria

Hypertension was defined as systolic blood pressure (SBP) ≥ 140 mmHg or diastolic blood pressure (DBP) ≥ 90 mmHg, or the prescription of medication for hypertension [23]. DL was defined as TC ≥ 5.18 mmol/L, and/or TG ≥ 1.7 mmol/L, and/or HDL-C < 1.04 mmol/L, and/or LDL-C ≥ 3.37 mmol/L, or use an antilipidemic medication [24].

### 2.5. Statistical Analyses

Analyses were performed using SPSS Version 23.0 (IBM, Chicago, IL, USA) and MedCalc Version 19.2 (MedCalc software Ltd., Ostend, Belgium). Initially, the basic characteristics of subjects were presented, and data were expressed as mean (standard deviation, SD) for normally distributed continuous variables and median (interquartile range, IQR) for not normally distributed continuous variables, whereas categorical variables were shown as counts (percentage). Then, differences between groups were assessed with Independent-sample *t* test for normally distributed continuous variables. Mann–Whitney U test was used for not normally distributed continuous variables, and categorical variables were assessed with the Chi-square test. Univariate logistic regression was applied to evaluate the risk factors for dyslipidemia. LAP and VCI were divided into four groups (Q1, Q2, Q3, Q4) for use in logistic regression because we observed no linear relationships between continuous independent variables and the logit transformed dependent variables. In order to better define the best model for explaining the result, the final model was produced via the enter method. Finally, the capacities of HPF indicators for predicting DL risk were compared via area under the receiver-operating characteristics (ROC) curves (AUC) analysis. All *p* values were two–sided and *p* < 0.05 was considered statistically significant.

## 3. Results

### 3.1. Subsection Differences in Anthropometry, HPF Indicators, and Blood Biochemistry

Details of the between-group differences in the HPF indicators, anthropometric and blood biochemistry characteristics of participants are shown in Table 2. A total of 776 university staff members were enrolled in this study, including 360 (46.39%) DL and 416 (53.61%) normolipidemic (NDL); the mean ages for DL and NDL were 45 and 38 years old, respectively. The prevalence of DL in males and females was 51.49% and 41.77%, respectively. However, no significant gender differences were observed (χ^2^ = 2.687, *p* = 0.101). Among the participants with DL, the prevalence of hypertension in males and females was 26.32% and 11.18%, respectively, while males had a higher prevalence of hypertension than females, and significant differences were evident by gender (χ^2^ = 9.104, *p* = 0.003). Compared with the NDL subjects, weight, BMI, BF, WHtR, LAP, SBP, DBP, GLU, TG, TC, and LDL-C were all significantly higher (*p* < 0.01), while VCI (*p* = 0.004), and SAR (*p* = 0.013) were significantly lower. However, height, SMI, GS, VG, and HDL-C did not differ between groups (*p* > 0.05).

### 3.2. Analyses of Risk Factors for Dyslipidemia

The results of the univariate logistic regression analysis are presented in Table 3. Males have a significantly higher risk of DL than females (OR = 1.480, 95CI%: 1.114–1.965). In addition, we found that the risk of DL was 1.292 times higher in hypertension than in normotensive subjects (OR = 2.292, 95CI%: 1.504–3.493). The risk of DL increased significantly with the increase in LAP levels in the fourth quartile as compared with the bottom quartile (OR = 9.846, 95CI%: 6.180–15.688). In terms of VCI, subjects in the fourth quartile had a 45.5% lower risk of DL (OR = 0.545, 95CI%: 0.364–0.816). Furthermore, increasing age, GLU, BMI, BF, and WHtR were significant risk factors for DL (*p* < 0.001), whereas SAR was negatively associated with the risk of DL (*p* = 0.012), and there was no correlation worth of attention between GS and VG (*p* > 0.05).

### 3.3. Diagnostic Accuracy of HPF Indicators for Predicting Risk of Dyslipidemia

The results of ROC curve analyses have been entered into Table 4. Overall, with the highest AUC for LAP, this factor was found to be significant in predicting DL (AUC = 0.730, 95CI%: 0.697–0.762, *p* < 0.001), with the best cut-off being 29.165, and it had a sensitivity of 48.7% and specificity of 87.6%. When grouped by gender, the best thresholds were 16.035 in males and 29.320 in females, respectively. As for females, the AUC (95CI%) values of BMI, BF, WHtR, and LAP were 0.512 (0.462–0.562), 0.548 (0.498–0.597), 0.567 (0.517–0.616), and 0.675 (0.627–0.721), respectively. The AUC of LAP was significantly greater than that of WHtR (Z = 5.479, *p* < 0.001). However, the AUCs for BMI and BF were not statistically significant (*p* > 0.05). In males, the AUCs corresponding to a 95CI% of BMI, BF, WHtR, and LAP were 0.649 (0.597–0.698), 0.573 (0.521–0.625), 0.682 (0.632–0.730), and 0.809 (0.765~0.848), respectively. The AUC of LAP was significantly higher than that of WHtR (Z = 5.421, *p* < 0.001), BMI (Z = 6.388, *p* < 0.001), and BF (Z = 6.845, *p* < 0.001), with a sensitivity of 70.6% and specificity of 81.1%. The ROC curves are presented in Figure 1.

## 4. Discussion

DL is a multifactorial disorder resulting from the interaction between genetic, environmental, and social factors [9,25]. Increasing evidence has shown that the prevalence of DL has increased continually over the past few decades, and the burden is also expected to increase in China [4,6]. In our present study, we found that female and male university staff members had high DL prevalence rates, and there was a significant relationship between HPF and DL risk. Furthermore, our study revealed the predictive value of LAP, BMI, WHtR, and BF for DL risk, and the results demonstrated that LAP showed better predictive ability than WHtR, BMI, and BF in both females and males. These findings would have relevance to the early prevention and screening interventions of DL for university staff.

### 4.1. Prevalence of DL among University Staff

With changes in socioeconomic development and lifestyle, the prevalence of DL is increasing in China [3]. A cross-sectional study of 65,128 subjects in Inner Mongolia showed that males (37.9%) had a significantly higher prevalence of DL than females (27.5%, *p* < 0.001) [6]. Another cross-sectional study of 14,744 members of the Chinese rural population in Henan showed that DL prevalence was higher in males (40.20%) than in females (35.92%) [4]. Our study revealed that male university staff (51.49%) had a higher prevalence than females (41.77%), much higher than previously reported. However, there was no statistically significant difference between genders (*p* > 0.05). This might be related to the high levels of education in our participants. Cho et al. [9] showed that biological and lifestyle risk factors are important in DL, which could be changed by education level; in parallel, education level had a significant impact on TC and LDL-C components. While the exact mechanisms by which the level of education affects DL remain unclear, they likely involve un-healthy lifestyles and increased psychosocial stress. According to a recent study, 88.8% of staff had moderate to high level of stress, and 25.1% slept less than 8 h nightly [26]. Some evidence suggested that exposure to stress was associated with an increased risk of diseases, including CVD, hypertension, obesity, and diabetes [27]. Other studies have found similar results, showing that those working in pressured environments (OR = 1.4) or with a history of work stress (OR = 3.2) were likely to develop CVD [28,29]. In accordance with literature, there were multiple other biological mechanisms by which work stress could trigger the molecular pathways leading to deleterious cardiovascular pathology, including increased secretion of cortisol and inflammatory cytokines (coagulation factor VII) [27,30,31]. On the other hand, a growing number of studies have shown that males had a higher rate of DL prevalence than females, whereas they exhibited lower rates of awareness, treatment, and control [3,4,32]. Health education and interventions for male university staff should be recommended in order to reduce the incidence of DL.

### 4.2. Risk Factors for DL in University Staff

Our results revealed that the risk of DL was associated with obesity-related body composition indicators (BMI, WHtR, LAP, BF), hypertension, males, age, and GLU. These results were consistent with those of previous studies [3,4,33]. It is well known that chronic over-nutrition combined with a lack of exercise, promotes the progression of obesity and metabolic diseases [34]. Hyperinsulinemia and insulin resistance were closely related to the development of metabolic disorder in obesity, and also a main cause of DL [35]. Another study showed that DL was associated with obesity, one of other relevant effects of insulin resistance and pro-inflammatory adipokines [36]. When a subject was insulin-resistant, the delayed clearance of TG-rich lipoproteins might result in HTG [36,37]. On the other hand, tumor necrosis factor-α (TNF-α) was highly associated with the development of DL and obesity-induced insulin resistance had already been addressed in previous studies. The induction of macrophages in adipose tissue produced pro-inflammatory cytokines such as interleukin-6 (IL-6) and interleukin-1 (IL-1), and further promoted inflammation [38,39]. In addition, it could increase oxidative stress through the activation of the NF-kappaB (NF-κB) pathway in adipose tissue [36,40].

Numerous studies have suggested that DL is associated with an increased risk of hypertension and CVD [41,42]. A cross-sectional study showed a positive correlation between DL levels and hypertension levels [43]. The fact that individuals with DL are liable to hypertension was attributed to many factors; for example, it affects the structure and function of the arteries, which could impair endothelial function, disrupt the production of nitric oxide and the regulation of blood pressure, and promote atherosclerosis [43,44]. Interestingly, adiposity-related confounders could be involved in the association between DL and risk of hypertension [43]. McGill et al. [45] observed that adipose tissue secreted high levels of leptin in obese individuals, thus inducing insulin resistance and the subsequent activation of the sympathetic nervous system and the renin-angiotensin system, which could result in elevated blood pressure. It is noteworthy that we observed similar pathological mechanisms in hypertension and DL. Therefore, controlling body weight and improving body composition, especially reducing body fat, could reduce the risk of DL as well as hypertension.

Our study results revealed that age and gender (males) were associated with an increased risk of DL, which was consistent with previous studies. The study found that age and gender were two important factors that could influence lipid levels. Levels of TC, LDL-C, and TG increased with aging, while HDL-C levels decreased, and the gender differences in lipid levels were further affected by age [46,47]. The reason was that visceral fat accumulation leads to a large amount of free fatty acids (FFA) and pro-inflammatory cytokines secreting from adipocytes and their associated macrophages, which further caused insulin resistance with age [48,49,50]. As far as gender is concerned, males are more likely to develop the disease than females. We believe that this difference could be due to the different ages of the study subjects. In the present study, the mean age of the females was 46.5 years and that of the males was 45.0 years. A retrospective study of over 230,000 individuals from 2009 to 2015 showed that the peak prevalence of DL in males was in the age group of 40–59 years, which was earlier than in females (peaked at 60–69 years) [6]. However, the exact mechanism of the augmented risk of DL in men compared with women is still unclear. 

Subsequently, in the present study, we also observed that GLU was s a risk factor for the development of DL. Qian et al. [51] revealed that GLU was positively correlated with SBP, DBP, TG, and TC, and was negatively correlated with HDL-C. Furthermore, a large number of studies have demonstrated that diabetes was an important independent risk factor for the development of CVD [52,53]. CVD was 2–4 times more prevalent in pre-diabetes patients with an impaired blood glucose level than in the normal population and resulted in a higher risk of DL [54,55]. This is because glucose metabolism affects lipid metabolism, as the latter depends on the former. It has been suggested that monitoring blood glucose level informed a clear age-related pathogenesis, which was important for CVD and DL prevention.

### 4.3. Relationship between HPF Indicators and Risk of DL

The results of this study showed that the level of CRF was significantly and negatively associated with the risk of DL. A longitudinal study with an approximately 6 years follow-up demonstrated that, with the development of fitness level, the risk of hypertension, metabolic syndrome, and HTG decreased by 28%, 52%, and 30%, respectively [56]. The current study also showed that CRF was a powerful predictor of future CVD events, and there was a significant negative correlation between them [14,57]. Another Chinese study of middle-aged women (40–49 years) showed that CRF was independently associated with the risk factors of CVD, including being overweight, hypertension, and DL [58]. This was consistent with the results of our research. A study with an 8.9 year follow-up found that the risk of developing atherogenic DL declined to 44% in those who maintained levels of CRF throughout the follow-up compared to those engaging in less CRF [59]. Furthermore, mounting experimental evidence suggests that, in addition to CRF levels, muscle strength was a predictor for CVD and CVD risk factors [14,60,61].

In this study, we did not observe the relationship between GS, VG, and the risk of DL without considering BMI. A study of 16,149 Japanese individuals aged 20–69 found that a higher relative GS (GS/BMI) and relative VG (VG/BMI) were associated with a lower incidence of DL [62]. Another study found that relative GS was inversely associated with the prevalence of DL and hyperlipidemia, as defined by Chinese guidelines [63]. The discrepancy in the results between studies could be explained by the different participants or the use of absolute versus relative grip strength. Momma et al. [62] believed that body size could influence the association between GS and DL, and that relative GS could be a risk marker of the development of DL. Additionally, we found SAR was also a risk factor for DL. Chen et al. [17] found that CVD risk factors were associated with flexibility and could offer a practical approach for predicting CVD risk. Another study demonstrated that flexibility was positively associated with HDL-C [64]. Furthermore, a Japanese study revealed that better trunk flexibility was negatively associated with DL prevalence [62].It has been reported that flexibility exercises reduce pro-inflammatory adipokines, including chemerin and Plasminogen activator inhibitor-1 (PAI-1), and increased anti-inflammatory adipokines, such as adiponectin [65]. In conclusion, flexibility exercises were essential for DL patients, and the improvement of CRF and flexibility could be useful for reducing the risk of DL.

### 4.4. Prognostic Value of Body Composition Indicators for Risk of Dyslipidemia

It was found that LAP, WHtR, BMI, and BF were valid predictors for the risk of DL. This has been confirmed in previous studies. Some evidence has indicated that visceral fat and metabolic disorder were linked to increased risk of CVD in patients with central obesity [66]. As is well known, BMI and WHtR are two common indexes of central obesity, and can facilitate the better assessment of CVD risk, but nothing else, given their inability to differentiate subcutaneous from visceral fat deposition. BMI used alone, could result in an underestimation of the adverse effects of obesity on CVD [67,68]. LAP was an emerging cardiovascular risk factor and is an index describing lipid over-accumulation based on WC and TG. It was strongly associated with visceral adiposity, and it could not only estimate CVD risk more effectively than BMI and WHtR, but also was significantly associated with all-cause mortality [69,70]. A 10-year CVD follow-up study of Caucasian adults reported LAP to be an independent predictor of CVD; the association of LAP with CVD remained significant, even after adjusting for hypertension, diabetes, physical activity, HTG, and pro-inflammatory cytokines (C-reactive protein, IL-6, TNF-α) [69]. At present, few studies have reported on the predictive value of LAP for DL, though the relevance of LAP to DL-related factors has been observed. A South Indian study reported that the ability to counteract insulin resistance was higher for LAP patients; further, ROC curve analysis showed that the area under LAP curve was 0.72, with a cut-off value of 33.4, and both the sensitivity and specificity were 75.0% [71]. Another study conducted with an Indian population showed that the LAP achieved the highest accuracy in predicting metabolic syndrome; the area under LAP curve was 0.901, the sensitivity was 76.4%, and the specificity was 91.1% [68]. Certainly, the studies in China the predictive ability of LAP has been validated, with high accuracy in elderly Chinese patients with intracranial atherosclerotic stenosis. It was also predictive for hypertension [72,73]. In our research, LAP was the best indicator for predicting DL, followed by WHtR. It has already been shown that WHtR could explain this difference in populations with different heights, with high specificity and sensitivity, and may be a better marker of CVD risk factors compared to BMI and WC [74,75]. A meta-analysis including 31 studies in 18 countries showed that WHtR could be a better prognostic and diagnostic marker for DL, hypertension, diabetes, and risk of CVD compared to BMI [76]. Another study demonstrated that WHtR was the best predictor of DL in southwest China, and the critical value is about 0.50 in females and males [77]. Similar results were also found in a Japanese study [78]. These results are consistent with our study (all: 0.494; female: 0.436; male: 0.504). Our findings also suggested that BMI was a valuable marker for predicting the risk of DL in male university staff members. However, the predictive accuracy of DL by BF was low in female and male university staff members. These research results showed that WHtR was strongly associated with DL or DL risk factors when the LAP was not being calculated. Which was easy to measure and inexpensive, and the diagnostic cut-off value was close between different ethnicities and genders. Therefore, WHtR can be used for large-scale epidemiological screenings in the early prevention and treatment of DL.

A major feature of the significance of our study was that it provided help for self-screening, assessment, and prevention of DL for university staff members. HPF indicators were convenient to use to measure and evaluate the risk of DL independently. However, some limitations of the present study should be taken into consideration. Firstly, the results of the study may not be generalized to other populations, as participants were university staff members. Secondly, our study could not determine the causal relationship between HPF and risk of DL due to its cross-sectional design. Finally, physical activity, lifestyle, and some physiological indicators were not included in the study. In the future, we would expand the sample size and longitudinal studies should be considered to verify the causal relationship between HPF and risk of DL. Additionally, the age-related mechanism of DL should be assessed.

## 5. Conclusions

In conclusion, our study found that the prevalence of DL was relatively high in university staff members, especially in males. The risk of DL was significantly related to body composition, cardiopulmonary fitness, and flexibility. It was worth noting that the LAP and WHtR performed better in predicting DL than BMI. It was suggested that we should focus on the prevention and screening of risk for DL in addition to medication. Particularly, HPF was significant for the prevention and control of DL, for example, by raising the CRF level, improving body composition, and promoting flexibility.

## Figures and Tables

**Figure 1 nutrients-14-00050-f001:**
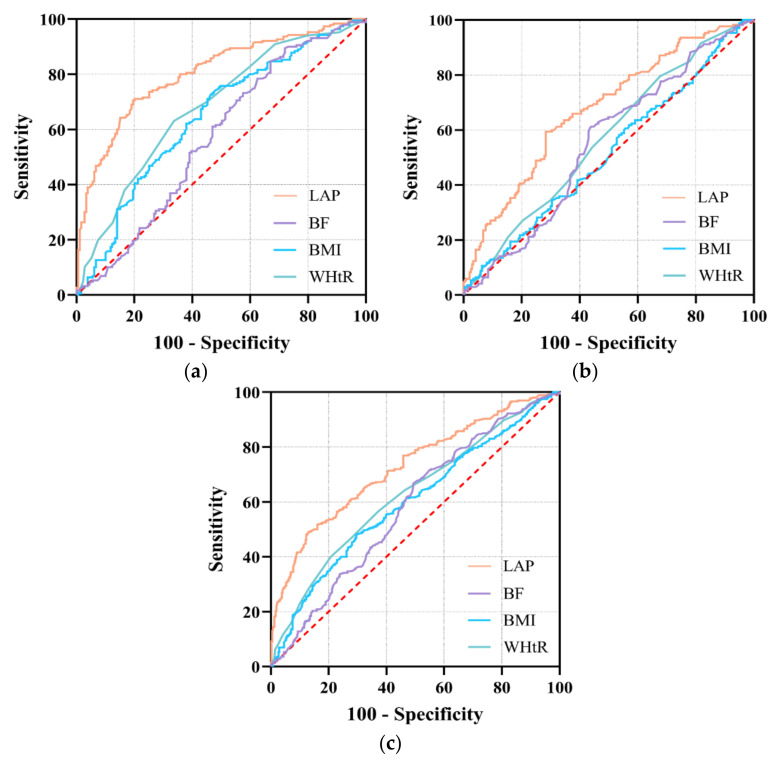
ROC curve of LAP, BF, BMI and WHtR for predicting dyslipidemia. (**a**): male; (**b**): female; (**c**): total; BMI: body mass index; BF: body fat; WHtR: Waist-to-height ratio; LAP: lipid accumulation product.

**Table 1 nutrients-14-00050-t001:** Participants characteristics (female/male).

Age (years)	40 (15)/42 (19)
Sex (*n*)	407/369
Height (cm)	1.62 (0.06)/1.73 (0.06)
Weight (kg)	56.70 (9.80)/73.20 (13.20)
BMI (Kg/m^2^)	21.79 (2.86)/24.53 (3.59)
WC (cm)	73.70 (8)/86.20 (9)

Data are represented by median (interquartile range, IQR); BMI: body mass index; WC: waist circumference.

**Table 2 nutrients-14-00050-t002:** Basic characteristic of the study participants.

Variables	Dyslipidemia (N = 360)	Normolipidemic (N = 416)	t/*χ^2^*/Z	*p*
Age (years)	45.00 (16.00)	38.00 (15.00)	−7.573	<0.001
Female	46.50 (15.00)	37.00 (13.00)	−8.069	<0.001
Male	45.00 (19.00)	40.00 (17.00)	−2.438	0.015
Dyslipidemia (*n*, %)	360 (46.39)	416 (53.61)		
Female (*n*, %)	170 (41.77)	237 (58.23)	2.687	0.101 ^a^
Male (*n*, %)	190 (51.49)	179 (48.51)		
Hypertension (*n*, %)				
Female (*n*, %)	19 (11.18)	4 (1.69)	9.104	0.003 ^b^
Male (*n*, %)	50 (26.32)	35 (19.55)		
Body composition				
Height (m)	1.68 (0.11)	1.66 (0.10)	−1.046	0.296
Weight (kg)	65.50 (18.03)	62.55 (15.00)	−3.203	0.001
BMI (kg/m^2^)	23.60 (4.33)	22.44 (3.61)	−4.723	<0.001
BF (kg)	18.30 (9.25)	16.75 (8.38)	−3.987	<0.001
WHtR	0.49 (0.07)	0.47 (0.04)	−5.994	<0.001
LAP	28.00 (27.25)	15.35 (14.22)	−10.919	<0.001
SMI (kg/m^2^)	8.49 (2.39)	8.69 (1.85)	−0.170	0.865
Muscle Fitness				
GS (kg)	35.25 (18.50)	32.60 (15.22)	−1.391	0.164
VG (cm)	35.56 (17.71)	35.75 (18.00)	−0.955	0.339
Cardiorespiratory fitness				
VCI (ml/kg) *	50.14 (11.71)	52.62 (12.26)	2.868	0.004
Flexibility				
SAR (cm)	7.48 (8.62)	9.00 (13.00)	−2.479	0.013
Blood pressure				
SBP (mmHg)	123.50 (18.00)	117.55 (16.00)	−5.804	<0.001
DBP (mmHg)	78.00 (13.00)	74.00 (12.00)	−5.500	<0.001
Blood biochemical				
GLU (mmol/L)	4.96 (0.57)	4.90 (0.51)	−3.362	0.001
TG (mmol/L)	1.37 (0.99)	0.87 (0.50)	−13.051	<0.001
TC (mmol/L)	5.45 (0.70)	4.45 (0.78)	−18.580	<0.001
HDL-C (mmol/L)	1.48 (0.56)	1.54 (0.38)	−1.913	0.056
LDL-C (mmol/L)	3.28 (0.68)	2.47 (0.63)	−17.054	<0.001

* Data are represented by mean (standard deviation, SD); ^a^: a chi-square test was conducted to compare the prevalence of dyslipidemia between different gender; ^b^: a chi-square test was conducted to compare the prevalence of hypertension between different gender; BMI: body mass index; BF: body fat; WHtR: waist-to-height ratio; LAP: lipid accumulation product; SMI: skeletal muscle mass index; GS: grip strength; VG: vertical jump; VCI: vital capacity index; SAR: sit-and-reach test; SBP: systolic blood pressure; DBP: diastolic blood pressure; GLU: glucose; TG: triglycerides; TC: total cholesterol; HDL-C: high-density lipoprotein cholesterol; LDL-C: low-density lipoprotein cholesterol.

**Table 3 nutrients-14-00050-t003:** Logistic regression model for risk factors associated with dyslipidemia.

Variables	β	SE	Wald	OR	95%CI	*p*
Gender						
Female	--	-	-	1.000	-	-
Male	0.392	0.145	7.331	1.480	1.114–1.965	0.007
Blood pressure						
Normotensive	-	-	-	1.000	-	-
Hypertension	0.829	0.215	14.885	2.292	1.504–3.493	<0.001
LAP						
Q1	-	-	-	1.000	-	-
Q2	0.414	0.222	3.485	1.513	0.980–2.338	0.062
Q3	0.886	0.218	16.603	2.426	1.584–3.717	<0.001
Q4	2.287	0.238	92.615	9.846	6.180–15.688	<0.001
VCI						
Q1	-	-	-	1.000	-	-
Q2	−0.165	0.203	0.660	0.848	0.569–1.263	0.417
Q3	−0.310	0.204	2.316	0.733	0.492–1.093	0.128
Q4	−0.607	0.206	8.669	0.545	0.364–0.816	0.003
Age	0.059	0.008	55.498	1.061	1.045–1.078	<0.001
GLU	0.649	0.159	16.730	1.914	1.402–2.613	<0.001
BMI	0.125	0.026	22.230	1.133	1.076–1.193	<0.001
BF	0.022	0.007	8.741	1.022	1.007–1.037	<0.001
WHtR	0.424	0.078	29.792	1.528	1.312–1.779	<0.001
SMI	0.021	0.043	0.243	1.021	0.939–1.111	0.622
GS	0.011	0.007	2.397	1.011	0.997–1.024	0.122
VG	−0.004	0.005	0.515	0.996	0.986–1.007	0.473
SAR	−0.020	0.008	6.295	0.980	0.964–0.996	0.012

BMI: body mass index; BF: body fat; WHtR: waist-to-height ratio; LAP: lipid accumulation product; SMI: skeletal muscle mass index; GS: grip strength; VG: vertical jump; VCI: vital capacity index; SAR: sit-and-reach test; GLU: blood glucose.

**Table 4 nutrients-14-00050-t004:** The comparisons of Health-Related Physical Fitness Indicators in predicting dyslipidemia risk.

Variables	Cut-Off Value	Sensitivity (%)	Specificity (%)	AUC (95% CI)	Z	*p*
Female						
BMI	18.765	94.0	10.6	0.512 (0.462–0.562)	6.036	<0.001
BF	16.750	60.2	56.6	0.548 (0.498–0.597)	4.577	<0.001
WHtR	0.436 *	77.7	35.3	0.567 (0.517–0.616)	5.479	<0.001
LAP	16.035 #	58.4	72.3	0.675 (0.627–0.721)		
Male						
BMI	23.955 #	73.3	53.7	0.649 (0.597–0.698)	6.388	<0.001
BF	14.950 *	84.5	33.7	0.573 (0.521–0.625)	6.845	<0.001
WHtR	0.504 #	63.1	66.3	0.682 (0.632–0.730)	5.421	<0.001
LAP	29.320 #	70.6	81.1	0.809 (0.765–0.848)		
All						
BMI	23.915 #	47.9	70.7	0.599 (0.563–0.634)	8.074	<0.001
BF	16.910 #	66.0	51.5	0.584 (0.548–0.619)	6.977	<0.001
WHtR	0.494 #	49.3	71.5	0.626 (0.590–0.660)	7.919	<0.001
LAP	29.165 #	48.7	87.6	0.730 (0.697–0.762)	-	-

* *p* < 0.05; # *p* < 0.001; BMI: body mass index; BF: body fat; WHtR: Waist-to-height ratio; LAP: lipid accumulation product.

## Data Availability

The data presented in this study are available on request from the corresponding author. The data are not publicly available due to confidentiality.

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
