# Peer review of "Association between Health-Related Physical Fitness and Risk of Dyslipidemia in University Staff: A Cross-Sectional Study and a ROC Curve Analysis"

_nutrients, 2021, doi:10.3390/nu14010050_

Round 1
Reviewer 1 Report
This article entitled " Association Between Health-Related Physical Fitness and Risk of Dyslipidemia in University Staff: A Cross-Sectional Study and a ROC Curve Analysis” has been proposed by Zhou et al and measure the role of Health-Related Physical Fitness and Risk in the context of Dyslipidemia.
The study is of potential interest but suffers from some minor errors and misrepresentations that need to be corrected
- The Abstract should be revised as it is too extensive;
- Several formatting and typing mistakes should be revised;
- English language needs improvement specially on Introduction and Discussion.
- I suggest the use subsections on the discussion;
- Do you have data on patient current medication? It would be highly relevant;
- Exclusion criteria: Age? What means physical dysfunction?
- Informed consent on personal medical data? Was it confidential?
- Potential ethical interest conflict as participants were members of the university staff.
Author Response
请参阅附件。

Reviewer 2 Report
This report describes the relationship between risk of dyslipidemia and health-related physical fitness. This aim was studied with 776 university staff members. The main finding was that the lipid accumulation product (LAP, derived from waist circumference and serum triglycerides) had the best relationship e.g. in ROC-analysis.
I have some comments and suggestions: The prevalence of dyslipidemia was about 46 %. It would be interesting to know what were the proportions of dyslipidemias due to 1. high LDL-C, 2. high TG, 3. combined dyslipidemia (high LDL-C + high TG) and 4. low HDL-C only. And - how does LAP work in different dyslipidemia types. The authors have the elements of metabolic syndrome. Therefore it would also be of interest to know, whether LAP works especially well in metabolic syndrome (using the harmonised criteria) as well.
The text reads well. Especially he discussion could be clearly shortened. I noticed a small typo in table 4. In column "Variables" Instead of "Meal" it should read "Male".
Round 2
Reviewer 1 Report
I do not have nothing else to add.
Reviewer 2 Report
Thank you for new and valuable analyses (tables and figures).
The new data shows that LAP is especially strong in prediction of DL, when DL is due to hypertriglyceridemia or combined hypertriglyceridemia because LAP is very strongly determined by TG-level. LAP is much less powerful, if DL is due to high LDL-C or low HDL-C. This is very important and must be stated in results and discussion.
Table 1 (response letter) shows that men more often have DL due to hypertrigyceridemia, which explains that LAP predicts DL better in men than in women. This is very important and must be stated in results and discussion.
Total prevalences of DL in table 1(response letter ) and Table 2(article) are not similar, why. In table 1(response letter) is about 34 % for women and 67 % for men (compared with 42 % and 51 % in table 2 (article) why ?
Interestingly, LAP was found to be a very good predictor of MetS (Fig 1, response letter), which has been found earlier e.g. in studies from India. This also is well understandable, because TG is a central component of MetS.
Minor: I found a new typo on page 9, line 12 "coculd", should be "could"